# Evaluating Vitamin D levels in Rheumatic Heart Disease patients and matched controls: A case-control study from Nepal

Lene Thorup[1,2]*, Sophie Amalie Hamann[2], Ashish Tripathee[3], Bhagawan Koirala[4], Bishal Gyawali[5,6], Dinesh Neupane[6,7], Cleonice C. Mota[8], Per Kallestrup[2], Vibeke E. Hjortdal[1,9]

1 Department of Cardiothoracic & Vascular Surgery, Aarhus University Hospital, Skejby, Aarhus, Denmark, 2 Department of Public Health, Center for Global Health (GloHAU), Aarhus University, Aarhus, Denmark, 3 Western Regional Hospital, Pokhara Academy of Health Sciences, Pokhara, Nepal, 4 Department of Cardiothoracic and Vascular Surgery, Manmohan Cardiothoracic Vascular and Transplant Center, Kathmandu, Nepal, 5 Department of Public Health, Section of Global Health, University of Copenhagen, Copenhagen, Denmark, 6 Nepal Development Society, Chitwan, Nepal, 7 Department of Epidemiology, Welch Center for Prevention, Epidemiology, and Clinical Research, Johns Hopkins Bloomberg School of Public Health, Baltimore, Maryland, United States of America, 8 Faculty of Medicine, Federal University of Minas Gerais, Belo Horizonte, Brazil, 9 Department of Clinical Medicine, Aarhus University, Aarhus, Denmark

* lenethorup@clin.au.dk

**Data Availability Statement:** All relevant data are within the paper and its Supporting Information files.

## Abstract

### Background

Diagnosis and treatment for Rheumatic Heart Disease (RHD) is inaccessible for many of the 33 million people in low and middle income countries living with this disease. More knowledge about risk factors and pathophysiologic mechanisms involved is needed in order to prevent disease and optimize treatment. This study investigated risk factors in a Nepalese population, with a special focus on Vitamin D deficiency because of its immunomodulatory effects.

### Methods

Ninety-nine patients with confirmed RHD diagnosis and 97 matched, cardiac-healthy controls selected by echocardiography were recruited from hospitals in the Central and Western region of Nepal. Serum 25(OH)D concentrations were assessed using dried blood spots and anthropometric values measured to evaluate nutritional status. Conditional logistic regression analysis was used to define association between vitamin D deficiency and RHD.

### Results

The mean age of RHD patients was 31 years (range 9–70) and for healthy controls 32 years (range 9–65), with a 4:1 female to male ratio. Vitamin D levels were lower than expected in both RDH and controls. RHD patients had lower vitamin D levels than controls with a mean s-25(OH)D concentration of 39 nmol/l (range 8.7–89.4) compared with controls 45 nmol/l (range 14.5–86.7) (p-value = 0.02). People with Vitamin D insufficiency had a higher risk (OR = 2.59; 95% CI: 1.04–6.50) of also having RHD compared to people with Vitamin D

**Funding:** LT received support to the study by the Frimodt-Heineke Fonden, http://frimodtheinekefonden.dk/, and Kong Christian den Tiendes Fond, http://kongehuset.dk/node/5556. VEH received funding from Lundbeckfonden (Grant R184-2014-2478), https://www.lundbeckfonden.com/en/. The funders had no role in study design, data collection and analysis, decision to publish, or preparation of the manuscript.

**Competing interests:** The authors have declared that no competing interests exist.

concentrations >50 nmol/l. Body mass index was significantly lower in RHD patients (22.6; 95% CI, 21.5–23.2) compared to controls (24.2; 95% CI, 23.3–25.1).

## Conclusion

RHD patients in Nepal have lower Vitamin D levels and overall poor nutritional status compared to the non-RHD controls. Longitudinal studies are needed to explore the causality between RHD and vitamin D level. Future research is also recommended among Nepali general population to confirm the low level of vitamin D as reported in our control group.

## Introduction

Rheumatic heart disease (RHD) affects an estimated 33 million people globally [1, 2], making it the most commonly acquired heart disease in people <25 years and almost as prevalent as human immunodeficiency virus [3]. Although the incidence of RHD in high-income countries decreased markedly during the 20th century, it remains a major public health concern in many low-and middle-income countries and marginalized communities in high-income countries–as such, three out of four children <15 years grow up in parts of the world where RHD is endemic [1].

RHD is characterized by chronic valvular lesions and is the result of Acute Rheumatic Fever (ARF), which develops as an autoimmune response to Group A Streptococcal (GAS) infection [4]. Currently, there is no curative treatment. The secondary prophylaxis consisting of benzathine penicillin G injections every 3–4 weeks is used to prevent recurrences, which can lead to new valvular lesions or to worsening of the previous one [5].

Despite knowing the etiologic agent, there are still many unanswered questions regarding pathophysiologic mechanisms involved in disease development and progression. The presence of residual autoreactive cells appears to play a role in the persistence and worsening of the valvular lesions in RHD patients [6, 7]. Not all GAS are rheumatogenic, and not all people are susceptible to developing ARF and RHD. While it is not known what makes a host susceptible [8], it is generally accepted that malnutrition has a great impact on the immune system affecting both innate and acquired immunity in children [9]. More specifically, hypovitaminosis D has been associated with an increased risk of infections such as GAS pharyngitis as well as the risk of developing autoimmune diseases [10, 11]. Links have also been demonstrated with ARFand rheumatic mitral stenosis [12, 13].

Finally, extravasation through the valvular endothelium seems to be an important step in the valvular lesions seen in RHD, and since s-25(OH)D regulates expression of vascular endothelial growth factor, this could explain the link between hypovitaminosis D and endothelial dysfunction, and subsequently ARF and RHD [14, 15].

Nepal is situated in a region with one of the highest prevalence rates of RHD in the world [2]. A school-based study estimated prevalence of subclinical RHD to be 10.2 per 1000 children [16]. Prevalence estimates for adults are not available from Nepal, despite peak prevalence usually occuring between 25–45 years of age. Nonetheless, RHD remains endemic in Nepal and the South-Asian region. Prevalence of insufficient Vitamin D levels in Nepal vary from 59.8% amongst new mothers to 17.2% in 6-8year olds [17, 18]. In healthy school-children in neighboring India, vitamin D deficiency prevalence is estimated to 35% [19]. To date, nutritional status in ARF/RHD patients in Nepal is unknown.

In this study we examined possible associations between s-25(OH)D levels and RHD status in Nepal. In addition, associations between socioeconomic factors and RHD is also investigated.

## Methods and materials

### Study design and area

This case-control study was carried out in the two largest cities in Nepal, Kathmandu and Pokhara, which are located in the Central and Western region of the country. Study participants were recruited from two hospitals; Western Regional Hospital (WRH) in Pokhara and Manmohan Cardiothoracic Vascular and Transplant Center (MCVTC) in Kathmandu, between March and July 2018. Both are governmental run health facilities, and have provided free prophylactic treatment to RHD patients through the national RHD control program funded by the Nepalese government from 2007 to 2018. MCVTC also provides free cardiac surgery for RHD patients, approximately 300 cases annually. The institutions are two of the largest governmental run health facilities in the country, and hence receive patients from all regions and districts.

### Subjects

Cases were selected from registries under the National ARF/RHD Prevention and Control Program when patients were seen for delivery of secondary prophylaxis, diagnostic and follow-up clinical evaluation or for surgical intervention. Patients were included based on a confirmed diagnosis of RHD by echocardiographic screening leading to registration in the national RHD Program Registry or new echocardiographic findings confirming RHD diagnosis, both in accordance with the World Heart Federation echocardiographic guidelines [5]. Exclusion criteria were issues which could interfere with Vitamin D metabolism or with the immunological conditions such as; patients below five years of age, recent hospitalization for more than one week, burn victims, chronic kidney disease and current tuberculosis or thyroid disease (self-reported).

Controls were selected by echocardiography and self-reported medical history among any people attending either institution as patients, relatives or acquaintants, matched on sex and age (maximun difference of 5 years). Exclusion criteria of controls were: diagnosis of ARF, RHD, congenital heart disease, any echocardiographic findings of valvular damage and otherwise the same exclusion criteria as for cases. Patients with congenital heart disease were excluded as they can display abnormal vitamin D concentrations [20, 21]. Cases were matched with controls from the same institution as themselves.

### Collection of socio-demographic information

Face-to-face interviews were conducted with all participants using a structured questionnaire containing questions on household items and living circumstances (S1 File). Socioeconomic status was assessed by computing a wealth index using principal components analysis. This divided the participant into terciles; poorest, middle and richest. The components used in the wealth index were; ownership of a house, animals, vehicles and electronic equipment (electricity, radio, television, mobile phone, telephone, refrigerator), furniture (bed, sofa, cupboard, computer, table, chair, clock, fan, dhiki/janto), housing characteristics and fuel used for cooking. Thus, this study measures SES based on durable assets ownership and access to utilities, to accommodate the often very fluctuating income patterns seen in many low and middle incomecountries, including Nepal.

A food frequency questionnaire was also included (S1 File). Participants were asked to tick the number of times they had consumed the most commonly available fruit and meat, fish, soybean oil, and egg in the last week. The purpose of the fruit intake was to detect differences in variance of diet between cases and controls, and thus give an idea of the general nutritional state of the two groups. Questions regarding meat, fish, soybean oil, and egg were added due to their possible direct effect on vitamin D concentrations and RHD development, making an assessment of differences between the two groups of interest. Questions used in this study are adapted from the Nepal Demographic and Health Survey [22] and Piryani et al. [23] to ensure cultural appropriateness.

## Anthropometric measurements

The following anthropometric measurements were determined: weight (kg), height (cm), age (years) and Mid-upper arm circumference (MUAC) (mm). The same weight scale and meter tape were used to measure cases and controls and measurements were performed by the same person. Shoes and heavy items of clothing were removed beforehand. Body mass index (BMI) was calculated as weight in kilogram divided by height squared in meter ($kg/m^2$). MUAC was measured on the non-dominant/left arm, except in patients with left-sided hemiplegia. Measurement was done at the mid-point between the olecranon and acromion on a relaxed arm. All participants were checked for pitting edema.

## Biochemical measurements

Blood was collected as dried blood spots on Perkin Elmer 226 Five Spot RUO Card filter paper, and extracted samples analyzed at the Department of Clinical Biochemistry, Aarhus University Hospital. We measured serum 25-hydroxyvitamin D2 (25(OH)D2, ergocalciferol) and serum 25-hydroxyvitamin D3 (25(OH)D3, cholecalciferol) using liquid chromatography-tandem mass spectrometry on Sciex Triple Quad 5500 LCMSMS System calibrated using 25-OH-Vitamin D3/D2 Serum calibration standards (ChromSystems). The method is adapted from Kvaskoff et al. [24], and S-25(OH)D is expressed as the sum of 25(OH)D2 and 25(OH) D3. Vitamin D deficiency was divided into three categories following Danish national standards, defined as: Vitamin D insufficiency (VDI) as a serum concentration between 25–50 nmol/L, moderate vitamin D deficiency (VDD) as a serum concentration of 13–25 nmol/L and severe Vitamin D deficiency as concentrations <13 nmol/L. For some analysis the term hypovitaminosis was used, defined as all concentrations < 50 nmol/l.

## Sample size

The prevalence of s-25(OH)D levels below 50 nmol/l was set to 77% in RHD patients based on previous studies [12]. Since data on vitamin D deficiency in Nepal is very inconsistent, we hypothesized hypovitaminosis D prevalence in non-RHD individuals to be 35%, based on results from a large study from Northern India [19]. We estimated sample size for two independent proportions with a 95% confidence level, 90% power and a margin of error (alpha) 5% to be 64; 32 cases and 32 controls.

## Statistical analysis

Data were collected in hard copies and entered using REDCap electronic data capture tools. Statistical analysis was performed using Stata Statistical Software IC 15.1 (StataCorp LP, TX). Two-tailed p-values $p \leq 0.05$ were considered statistically significant.

Normality in distribution was tested using q-q plots. Normally distributed continuous variables were compared between groups using Student's t-test and reported as means with standard deviation (SD). Differences between groups of categorical variables were compared using Chi-square test. SES was calculated by creating wealth index scores for each participant using principal components analysis (PCA), following the *Measuring Equity with Nationally Representative Wealth Quintiles* guide [25]. Due to the smaller sample size in this study compared to large epidemiological surveys, quintiles were converted into terciles. Multivariate conditional logistic regression analysis between sufficient Vitamin D concentrations and hypovitaminosis D were performed, adjusting for potential confounders; age, sex, BMI, education, and SES [26–32]. Potential confounders were identified a priori through a literature review. Paired 1:1 matching between cases and controls was performed after data collection, except 5 cases and 1 control who were 2:1 matched, because of the post hoc exclusion of 12 subjects (Fig 1). A sensitivity analysis of available variables before and after excluding the 12 subjects revealed no difference in major (or other) variables.

## Ethical considerations

This study received ethical approval from the ethical review board of the Nepal Health Research Council (ref. no. 2398). Written information was handed out to participants and relatives in Nepal's official language Nepali. Prior to data collection, an informed consent form had to be signed. If illiterate, oral information was given and fingerprint was used as signature. Informed assent form was obtained from participants under 18 years of age. After biochemical analysis, all results were reported back to the participants along with advice of treatment if needed. This was done by local health personnel in Nepali language.

## Results

A flowchart illustrating exclusions for the different steps in data analysis is illustrated in Fig 1. The study excluded 6 cases because they were misclassified as having RHD when in fact their echocardiography report revealed no significant disease. Later, an additional 6 people were excluded before any analysis regarding s-25(OH)D concentrations were performed because these 6 had comorbidities assessed to potentially affect their s-25(OH)D concentrations. They could still be part of SES calculation, dietary intake, and other demographic measures since their comorbidities would not influence these parameters.

In total, 99 patients with confirmed RHD diagnosis and 97 matched controls were included from two different sites in Nepal. A comparison of unadjusted sociodemographic, anthropometric and socioeconomic differences is presented in Table 1.

The mean age of all participants was 31.71 years ± 11.5 and 75.5% were female. There was no significant difference between cases and control on gender and age.

BMI was significantly lower in cases than in controls, 22.4 ±4.5 kg/m$^2$ and 24.2 ±4.6 kg/m$^2$, respectively. Mean MUAC for cases were 253 ±34 mm and 266 ±43 mm for controls.

Almost half (47%) of cases belonged to the lowest possible SES class. For controls, this number was 21%. In the highest SES class, 25% of cases were represented compared to 40% of controls. The cases did not differ significantly on any other sociodemographic variables including overcrowding, compared to their controls.

We observed VDI and VDD in 78% of cases and mean concentration was significantly lower compared to controls. The prevalence of VDI and VDD in controls were 70%. Fig 2 displays the distribution of s-25(OH)D concentrations in all groups.

Univariate conditional logistic regression analysis revealed only association between Vitamin D deficiency (<25 nmol/l) and RHD (OR = 3.14; 95% CI: 1.02–9.64) but not for Vitamin

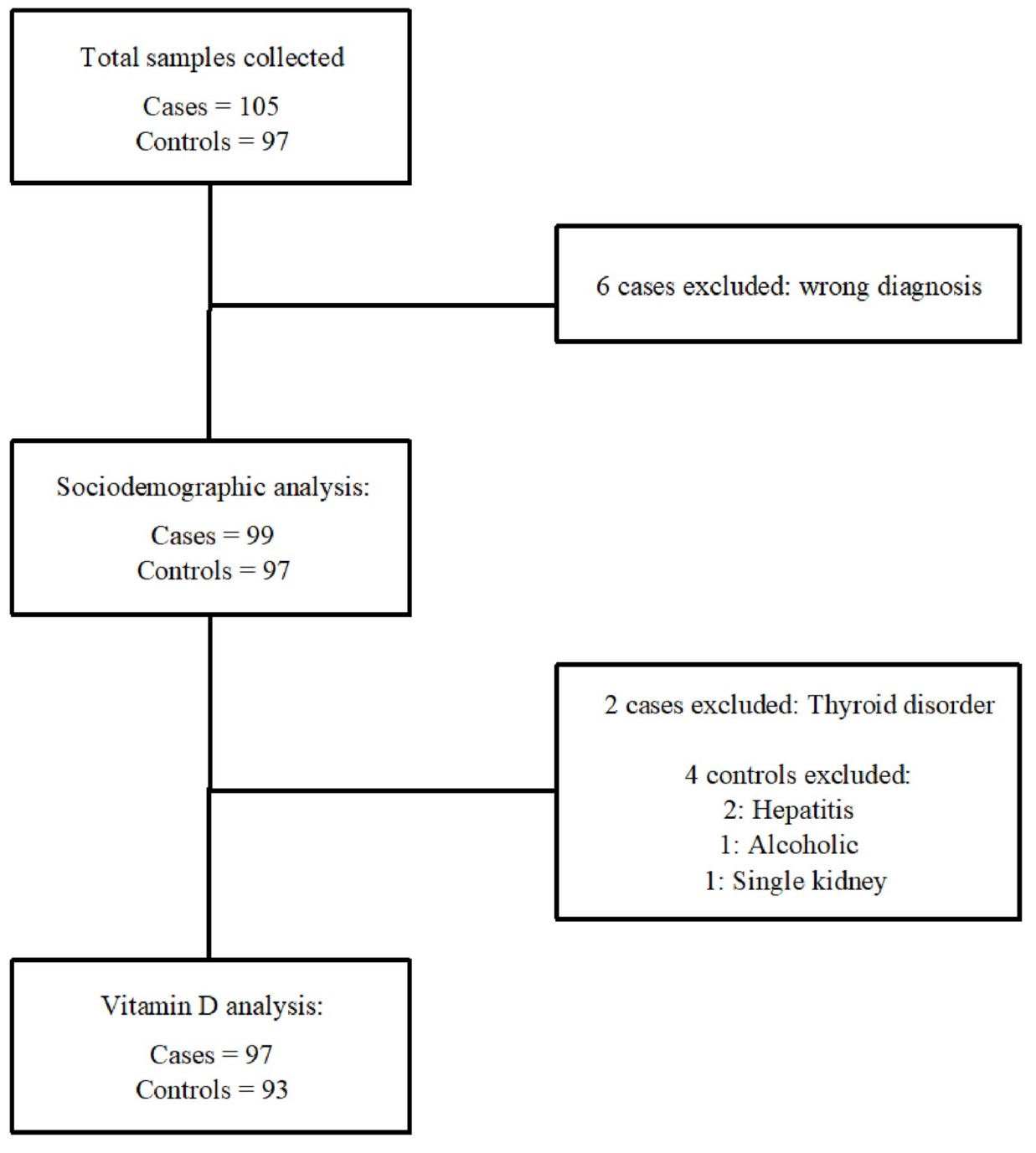

**Fig 1. Flowchart of inclusions and exclusions.**

D insufficiency. However, after adjusting for potential confounders, multivatiate analysis found a significant association (OR = 2.59; 95% CI: 1.04–6.50) for Vitmain D insufficiency but a non significant association for Vitamin D deficiency (Table 2).

Detailed echocardiography report was available from 41 patients. Upon Chi-squared test of the report from the 41 patients available, there was no correlation between the disease stage

**Table 1. Sociodemographic characteristics of cases and controls.**

| | RHD Patients [n = 99] | Controls [n = 97] | P for difference |
|---|---|---|---|
| Female % | 77% | 71% | 0.500 |
| Age (y), n (%) | | | |
| 5–19 | 13 (13) | 12 (12) | |
| 20–29 | 28 (28) | 28 (29) | |
| 30–39 | 40 (41) | 36 (37) | |
| 40–49 | 10 (10) | 12 (12) | |
| >50 | 8 (8) | 9 (10) | |
| | | | 0.976 |
| Mean age (y) ±SD | 31 ± 11.4 | 32 ± 11.6 | 0.700 |
| Mean BMI (kg/m$^2$) ±SD | 22.2 ± 4.4 | 24.2 ± 4.6 | 0.002 |
| Mean MUAC mm ± SD | 252 ± 34 | 266 ± 43 | 0.013 |
| Socioeconomic status, n (%) | | | |
| Poorest | 46 (47) | 20 (21) | |
| Middle | 27 (28) | 38 (39) | |
| Richest | 25 (25) | 39 (40) | |
| | | | 0.001 |
| Number of siblings mean ± SD | 4.2 ± 2.4 | 3.8 ± 2.1 | 0.300 |
| Sisters | 2.3 ± 2.0 | 1.9 ± 1.5 | 0.120 |
| Brothers | 1.8 ± 1.4 | 1.9 ± 1.4 | 0.770 |
| Sleeping arrangement | | | |
| Bed | 83 | 82 | |
| Blanket/mattress on floor | 14 | 15 | |
| | | | 0.959 |
| People sleeping in same room (overcrowding) | | | |
| 1–2 | 14 | 15 | |
| 3–6 | 82 | 79 | |
| 7–10 | 2 | 3 | |
| | | | 0.867 |
| Heart disease in family | 11 | 11 | |
| RHD | 5 | 5 | |
| CHD | 2 | 0 | |
| Other | 4 | 6 | |
| | | | 0.301 |
| ARF in family | 10 | 15 | 0.590 |
| Family type | | | |
| Nuclear | 29 | 34 | |
| Joint | 61 | 60 | |
| | | | 0.789 |
| Mean s-25(OH)D (range) [n] | 38.7 (8.7–89.4) [97] | 44.7 (14.5–86.7) [93] | 0.014 |

Characteristics of 99 rheumatic heart disease patients and 97 controls. SD: Standard Deviation; BMI: body-mass-index, kg/m$^2$; MUAC: middle-upper-arm circumference, mm; RHD: rheumatic heart disease; CHD: congenital heart disease; RF: rheumatic fever.

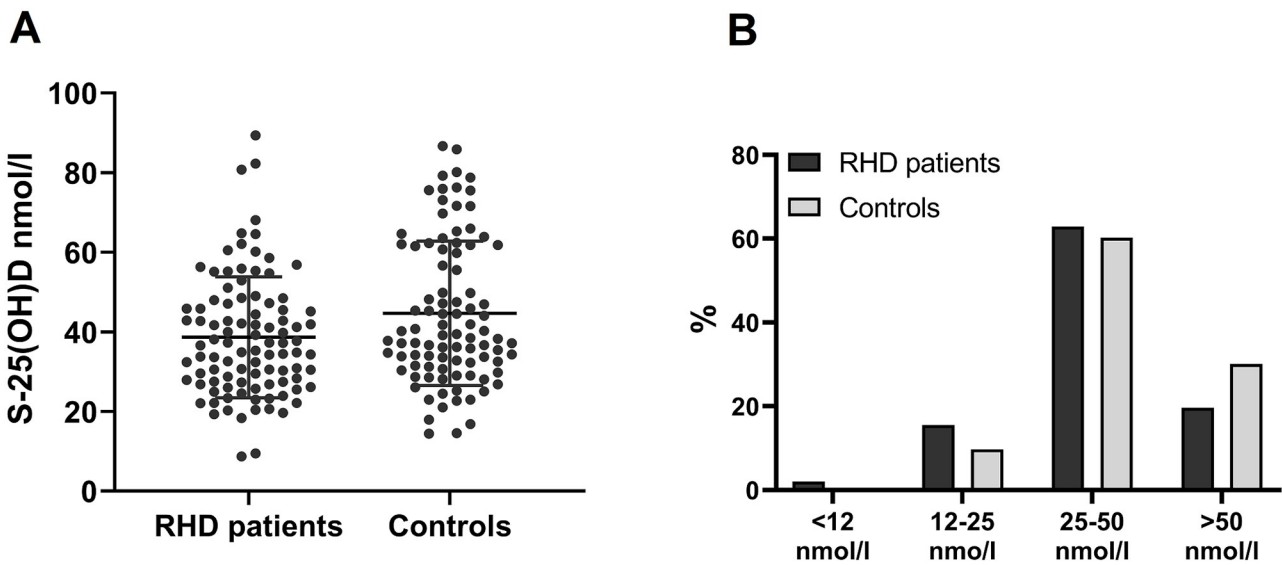

**Fig 2. Distribution of s-25 (OH)D concentrations.** (A) Distribution of s-25(OH)D concentrations in cases and controls with mean and standard deviation. (B) Distribution of rheumatic heart disease patients and healthy controls with severe Vitamin D deficiency (<12 nmol/l), Vitamin D deficiency (12–25 nmol/l), Vitamin D insufficiency (25–50 nmol/l) and sufficient (>50 nmol/l) s-25(OH)D levels.

and lower s-25(OH)D concentrations. The same was applicable when the patients without detailed echocardiographic reports were distributed into the s-25(OH)D categories considered sufficient, VDI and VDD and showed the same pattern of distribution.

Analysis of the food-frequency questionnaire revealed no statistically significant differences in intake between cases and controls (Table 3).

## Discussion

Both RHD patients and controls in Nepal have a high prevalence of vitamin D deficiency and insufficiency. RHD patients on average have a significantly lower s-25(OH)D concentration compared to their controls. Especially, the prevalence of VDD was higher in RHD patients. This is to date the largest study on Vitamin D and RHD. Three other studies have examined the relationship between vitamin D concentrations and RF or RHD [12, 13, 33]. The previous studies showed a 27–59% lower concentration in patients whereas we only demonstrated a

**Table 2. Odds ratio of rheumatic heart disease diagnosis in relation to s-25(OH)D status.**

| | Unadjusted | | | Adjusted[a] | | |
|---|---|---|---|---|---|---|
| | OR | 95% CI | P value | OR | 95% CI | P value |
| s-25(OH)D >50 nmol/l | Reference | | | Reference | | |
| s-25(OH)D 50–25 nmol/l | 1.61 | 0.78–3.33 | 0.20 | 2.59 | 1.04–6.50 | 0.04* |
| s-25(OH)D <25 nmol/l | 3.14 | 1.02–9.64 | 0.05* | 3.62 | 0.93–14.09 | 0.06 |

* Statistically significant *p-values*.

[a] Adjusted for: age, BMI, sex, education, and SES.

Unadjusted (left) and adjusted (right) Odds Ratio of RHD diagnosis in relation to s-25(OH)D status. Sufficient Vitamin D status used as a reference value.

**Table 3. Dietary intake in the last week stratified by cases and controls.**

|  | RHD patients | Controls | P value |
|---|---|---|---|
| **Fruit intake, n (%)** |  |  |  |
| Not consumed | 19 (19) | 11 (11) |  |
| 1 time/week | 7 (7) | 14 (15) |  |
| 2–6 times/week | 47 (48) | 46 (47) |  |
| > 7 times/week | 26 (26) | 26 (27) |  |
|  |  |  | 0.216 |
| **Meat, n (%)** |  |  |  |
| Not consumed | 10 (10) | 7 (7) |  |
| 1 time/week | 37 (38) | 35 (36) |  |
| 2–6 times/week | 44 (45) | 46 (47) |  |
| > 7 times/week | 7 (7) | 9 (10) |  |
|  |  |  | 0.832 |
| **Egg, n (%)** |  |  |  |
| Not consumed | 30 (31) | 27 (28) |  |
| 1 time/week | 26 (26) | 19 (20) |  |
| 2–6 times/week | 32 (33) | 41 (42) |  |
| > 7 times/week | 10 (10) | 10 (10) |  |
|  |  |  | 0.503 |
| **Soybean Oil, n (%)** |  |  |  |
| Not consumed | 44 (45) | 40 (41) |  |
| 1 time/week | 8 (8) | 3 (3) |  |
| 2–6 times/week | 19 (19) | 18 (19) |  |
| > 7 times/week | 27 (28) | 36 (37) |  |
|  |  |  | 0.287 |
| **Fish, n (%)** |  |  |  |
| Not consumed | 59 (60) | 62 (64) |  |
| 1 time/week | 30 (31) | 25 (26) |  |
| 2–6 times/week | 9 (9) | 10 (10) |  |
| > 7 times/week | 0 (0) | 0 (0) |  |
|  |  |  | 0.750 |

Dietary intake in last one week, in rheumatic heart disease patients and controls.

12.9% difference. However, Yusuf et al. only included RHD patients, no healthy controls, and only looked at calcification (Wilkins calcium score) of exclusively stenotic mitral valves [33]. Onan et al. only included patients with rheumatic mitral stenosis [13] and the third study only included ARF patients and none with RHD [12]. Since inclusion criteria vary between studies, it is not surprising that results also vary. Worth noting is the fact that they all have the same conclusion: s-25(OH)D is significantly lower in patients with RHD. However, it is not possible to report on the causality of this relationship. Vitamin D status can change throughout time, making it difficult to say if the lower concentration was present before the patient developed RHD. For instance, the chronically ill RHD patient may not be active and going outside as much as before falling ill, thus being less exposed to sunlight which might explain a lower s-25 (OH)D concentration. To address these questions a longitudinal study is required. Unfortunately, such data is not available from the study area, and current status is here accepted as a compromise.

To make sense of these findings, an understanding of the pleiotropic nature of Vitamin D is important. Vitamin D receptors (VDR) and the activating enzyme 1α-hydroxylase are identified in over 30 target organs including many cells of the immune system [34]. In the innate immune system, vitamin D enhances macrophages' phagocytic abilities and monocytes increase expression of 1α-hydroxylase and VDR through toll-like receptor signaling when encountering pathogens. The VDR complexes, in turn, activate transcription of antimicrobial cytokines [10]. Vitamin D also affects the adaptive immune system by modulation of antigen presenting cells to a more immature state, expressing fewer major histocompatibility complex class II molecules and thereby presenting fewer antigens and producing less interleukin-12, resulting in reduced activation of B and T cells. Furthermore, the modulated antigen presenting cells produce more interleukin-10—a tolerogenic cytokine [35]. Altogether, Vitamin D enhances the body's resistance against infections while decreasing the risk of an inappropriate autoimmune response. In general, RHD patients to a greater extent suffer from malnutrition with both lower BMI and MUAC. Whether they were malnourished before becoming ill or malnourished because they were ill, is not possible to determine from this study. It could be part of a two-way causal relationship as undernutrition lower immune responses, thereby increasing the risk of infection, and infection consequently aggravating undernutrition by an increase in demands of nutrients while simultaneously decreasing appetite in the affected individual [9].

Overcrowding and social disadvantage are two of the most commonly reported risk factors for GAS infection leading to the first espisode of ARF in susceptible individuals as well as resulting in recurrences in RHD patients [36]. This study also demonstrated a clear association between low SES and RHD but did not find an association with overcrowding. This could be because the mean age of patients was 31 years. Since the first acute episode of ARF occurs more frequently in the age group of 5–14 years [37], this means their housing situation could have changed since their childhood and adolescent years when overcrowding contributed for a higher risk of exposure to GAS and triggered ARF in those susceptible individuals. On the other hand, in Nepal, it is not unusual to live as a joint family throughout life, as demonstrated in Table 1.

We found slightly higher s-25(OH)D concentrations in samples collected from MCVTC compared to samples from WRH. However, geographical plottings showed that participants from both institutions represent both central and peripheral districts, and the difference between patients and controls was still present when comparing the groups within the two institutions.

No association was found between dietary intake and RHD (Table 3). This is in contrast to the general acceptance of fish and meat products as an important source of Vitamin D [38], as well as previous studies where consumption of soybean oil and egg, has been shown to suppress the rheumatic process because of their high concentrations of phospholipids and palmitamide [39].

Finally, we would like to highlight that the majority of RHD patients were young women. Though not surprising, it is still an important aspect that raises concern especially since the age of the women coincides with childbearing age. Having a heart disease, including RHD, whilst becoming pregnant can lead to serious complications [40]–particularly when access to health care is limited. In fact, RHD has been suggested a leading cause of indirect obstetric death in some sub-Saharan countries [30]. Yet no one has been able to explain why women are more often affected by RHD than men [27, 28, 30]. It is worth noting that this difference is not present when comparing ARF prevalence. Explanations to this sex-based difference could include extrinsic factors such as reduced access to primary and secondary ARF prophylaxis for girls and culturally rooted disadvantages affecting female health in general. Additionally,

intrinsic factors increasing host susceptibility such as immunogenic, genetic and in particular hormonal differences, should also be considered and investigated further.

This study has some limitations. The major limitation being the usage of current vitamin D status as a proxy for previous concentrations. Furthermore, the higher than expected prevalence of hypovitaminosis D in the background population reduces the power of the study. However, the impact was limited by including 3 times as many participants than required in the sample size calculations. Nonetheless, the sample size is relatively small. This should also be considered when understanding the measurement of associations. While some are statistically significant, they are largely marginal. This reduces the immediate clinical applicability of the results, but should be seen in light of the aforementioned smaller sample size. Increasing the sample size for future studies could help make the associations clearer.

The dried blood spot sampling procedure for vitamin D concentration was validated twice–once before and once after data collection—at Aarhus University Hospital, using the exact same method as in this study, hence the sampling procedure should not be a considerable source of error. Furthermore, all anthropometric data were measured upon inclusion and not self reported or taken from previous examinations, adding value to the results.

However, the lack of systematic record keeping in the Nepalese health system made identification of comorbidities a challenge. All participants were questioned about their medical history and whenever possible, their record books were examined thoroughly, but on many occasions, the patients did not bring their record books, which could result in underreporting of comorbidities that might affect s-25(OH)D concentrations. Additionally, there is a risk of recall bias when conducting a survey, especially the food frequency intake should be interpreted with caution.

In light of the abovementioned limitations and the very nature of the study design, we suggest caution when generalizing to other populations, but instead advise more research in the field of vitamin D deficiencies especially in low income countries, and how it affects immune responses.

## Conclusion

Both Vitamin D Insufficiency (VDI) and Vitamin D Deficiency (VDD) is highly prevalent among Rheumatic Heart Disease (RHD) patients in Nepal. RHD patients presented with significantly lower mean s-25(OH)D concentrations and overall poor nutritional status compared to the non-RHD controls. People with Vitamin D insufficiency had a higher risk (OR = 2.59) of also having RHD, underlining the potential of hypovitaminosis D being either a risk factor or feature of RHD, but longitudinal studies are needed to explore the causality of this relationship further. Larger studies among the Nepali population is also recommended to confirm the high prevalence of hypovitaminosis D found in our control group.

Other potential risk factors found in this study include low BMI, low mid upper arm circumference, low socioeconomic status, and female sex.

## Supporting information

**S1 Checklist. STROBE checklist.**
(DOCX)

**S1 File. Questionnaire used to collect information on demographic variables and for wealth index score analysis.**
(DOCX)

**S1 Dataset. Minimal data set.**
(XLS)

## Acknowledgments

The authors are grateful for the support from Nepal Development Society in planning and carrying out this project. Furthermore, we would like to thank the clinical staff and students at Western Regional Hospital, Pokhara and Manmohan Cardiothoracic Vascular and Transplant Center, Kathmandu, laboratory staff at Aarhus University Hospital and Mr. Kishor Pandey from Everest International Clinic and Research Center for consultancy.

## Author Contributions

**Conceptualization:** Lene Thorup, Bhagawan Koirala, Bishal Gyawali, Dinesh Neupane, Per Kallestrup.

**Data curation:** Lene Thorup.

**Formal analysis:** Lene Thorup, Bishal Gyawali, Dinesh Neupane, Vibeke E. Hjortdal.

**Funding acquisition:** Lene Thorup, Vibeke E. Hjortdal.

**Investigation:** Lene Thorup, Sophie Amalie Hamann, Ashish Tripathee, Bhagawan Koirala.

**Methodology:** Lene Thorup, Sophie Amalie Hamann, Bishal Gyawali, Dinesh Neupane, Per Kallestrup, Vibeke E. Hjortdal.

**Project administration:** Lene Thorup, Per Kallestrup, Vibeke E. Hjortdal.

**Resources:** Lene Thorup, Vibeke E. Hjortdal.

**Supervision:** Ashish Tripathee, Bhagawan Koirala, Bishal Gyawali, Dinesh Neupane, Per Kallestrup, Vibeke E. Hjortdal.

**Validation:** Lene Thorup.

**Visualization:** Lene Thorup, Per Kallestrup.

**Writing – original draft:** Lene Thorup.

**Writing – review & editing:** Lene Thorup, Sophie Amalie Hamann, Ashish Tripathee, Bhagawan Koirala, Bishal Gyawali, Dinesh Neupane, Cleonice C. Mota, Per Kallestrup, Vibeke E. Hjortdal.

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
