## [Decision Letter · Decision Letter 0]

30 Mar 2020

PONE-D-20-02934

Evaluating Vitamin D levels in Rheumatic Heart Disease Patients and Matched Controls: A Cross-sectional Study from Nepal

PLOS ONE

Dear Ms. Thorup,

Thank you for submitting your manuscript to PLOS ONE. After careful consideration, we feel that it has merit but does not fully meet PLOS ONE’s publication criteria as it currently stands. Therefore, we invite you to submit a revised version of the manuscript that addresses the points raised during the review process.

Despite the claim that the cases and controls had been age-matched, the age distribution provided in Table-1 shows considerable discrepancy, especially in the age groups 15-24 and 25-34. This should be convincingly explained by the authors.It is good that the authors attempted to control for some variables (age, BMI, sex, education, and SES) in their conditional logit regression model. However, adequate information had not been given how these variables had been selected for adjustment. The decision to adjust the matching variable “age” also need to be explained. The criteria for selecting variables for adjustment should be very convincing for the readers because the association between the VD and RHD status only becomes significant after the adjustment.One of the major limitations of the study is that the vitamin D status of the study subjects was determined many years after the outcome of interest had happened. And this has not been acknowledged and discussed in the manuscript. Do you think that the current vitamin D status can be considered as a proxy indicator for the pre RHD vitamin D status?Please present the association of RHD both with Vitamin D Insufficiency and Vitamin D Deficiency so that readers could have comprehensive understanding on the relationship between the two variables.The sample size calculation appears to be confusing. What specific formula did you use for the determination of sample? What was the expected prevalence in the RHD subjects?The purpose of the ROC analysis in the study is not clear and I recommend the authors to drop this analysis and related discussions and conclusions from this paper.Please consider a sensitivity analysis to evaluate whether the decision to exclude the 12 subjects has significantly affected the conclusion of the study or not.As most of the study subjects are adults, some of the variables of the study including maternal education, maternal occupation, and type of school are less relevant and I kindly recommend you to remove them. I also don’t understand the purpose of presenting HFA standard curves for boys and girls less than the age of 19 years while most of the study subjects are actually adults.The variables of the study (apart from RHD and VD status) should be clearly listed and described in the methods section.As suggested by the second reviewer, the study should be described as a case-control study.

We would appreciate receiving your revised manuscript by Apr 27 2020 11:59PM. To enhance the reproducibility of your results, we recommend that if applicable you deposit your laboratory protocols in protocols.io, where a protocol can be assigned its own identifier (DOI) such that it can be cited independently in the future. For instructions see: http://journals.plos.org/plosone/s/submission-guidelines#loc-laboratory-protocols

We look forward to receiving your revised manuscript.

Kind regards,

Samson Gebremedhin, PhD

Academic Editor

PLOS ONE

Additional Editor Comments (section-by-section comments):

Abstract

Is this sentence really relevant for the research question posed? “Diagnosis and treatment for Rheumatic Heart Disease (RHD) is inaccessible for many 26 of the 33 million people in low- and middle-income countries living with this disease” 

Background

Line 94-95: Can you rephrase the phrase “occurrence of RHD” to “RHD status” because the latter is more compatible with cross-sectional and case-control designs

Methods

Line 118: please provide your matching criteria for age. i.e. the age range considered for matching of cases and controls

Line 154: Confidence interval > confidence level

Results

Table 1: “Less poor” > “middle”

Table 1: The variable “Type of school” is not clear

Table 1: Please redefine age ranges based on wider intervals so that statistical test would be feasible.

Line 236-237: “Analysis of the food-frequency questionnaire revealed no statistically significant differences in intake 237 between cases and controls”. The type of the food frequency questionnaire should be described in the methods section. 

Discussion

I recommend the authors to integrate the section “strength and limitation” with the discussion section.

Conclusion

Line 340: “Vitamin D status could not be used as a predictor of RHD diagnosis”. To start with, was there any priori hypothesis to assume that the current vitamin D status could be used as a predictor of RHD that happened many years back?

Journal Requirements:

Reviewers' comments:

Reviewer's Responses to Questions

**Comments to the Author**

1. Is the manuscript technically sound, and do the data support the conclusions?

Reviewer #1: Yes

Reviewer #2: Yes

2. Has the statistical analysis been performed appropriately and rigorously? 

Reviewer #1: Yes

Reviewer #2: Yes

3. Have the authors made all data underlying the findings in their manuscript fully available?

Reviewer #1: Yes

Reviewer #2: Yes

4. Is the manuscript presented in an intelligible fashion and written in standard English?

Reviewer #1: Yes

Reviewer #2: Yes

5. Review Comments to the Author

Reviewer #1: A well written manuscript which is clinically relevant. Please address the following :

1. Please explain the categories of the socioeconomic status : poorest, less poor and richest.

2. The data on hours of TV per day is redundant and sounds unscientific. Kindly remove or explain its relevance.

3. Please remove the figures on height for age in girls and boys. These are not part of the study objectives.

Reviewer #2: Thank you for this well designed and written study. I have only some minor revisions for the authors to consider.

1. The title refers to both a case control and cross section study. In my view it qualifies as a case control study and so could be simply called this. It is a point in time study, yes, but this is the nature of C-C studies where we start with the known outcome and compare exposures retrospectively. The authors could reconsider the title.

2. Minor text edits

27-28: in order to prevent and optimise treatment. Could be made clearer that it is to prevent disease and optimise treatment.

37-38: suggest slight change to wording with addition of 'compared with' controls instead of 'while for'

43: add s to control

77: develop-ing

91-92: I am not sure I understand what is meant by 'Vitamin D status vary among from 59.8%'. Could you clarify what this measurement is? e.g. low status, known status.

116: for TB and thyroid disease, was it both current and previous disease excluded or just current?

119: I am interested in why congenital heart disease patients were excluded. Other studies on RHD specifically use this population for controls. MAybe a sentence to justify this would be useful for others considering designing similar studies on RHD to help them understand your decision.

Table 3: Woudl be good to include the % as well as in Table 1.

3. Age groups: if they were age matched, why is there a discrepancy in the age group numbers? Maybe it needs to be better explained in the methods section how you age matched so readers can understand this better.

4. Nutritional differences: while no differences were found among males, the respective sample sizes were small. It could then be that you simply didn't have the power to detect a true difference among the males. This could be discussed more in your limitations section.

5. 310-311: I think you need to add in references to other studies here that support your claims with regards to preponderance of females with RHD.

6. PLOS authors have the option to publish the peer review history of their article (what does this mean?). If published, this will include your full peer review and any attached files.

Reviewer #1: Yes: Sakthiswary Rajalingham

Reviewer #2: No

---

## [Author Response · Author response to Decision Letter 0]

27 Apr 2020

Dear Dr. Samson Gebremedhin and reviewers at PLOS ONE

Thank you for your comments on our manuscript PONE-D-20-02934 entitled “Evaluating Vitamin D levels in Rheumatic Heart Disease Patients and Matched Controls: A Cross-sectional Study from Nepal”. We have read your comments with great interest and revised our manuscript accordingly. As requested a version of the manuscript with marked changes as well as a version without marks have been uploaded alongside this rebuttal letter. 

Comments from editor and reviewers:

Editor

- Despite the claim that the cases and controls had been age-matched, the age distribution provided in Table-1 shows considerable discrepancy, especially in the age groups 15-24 and 25-34. This should be convincingly explained by the authors.

RESPONSE: We fully agree that the intervals set display discrepancies and have corrected accordingly with more appropriate cutoffs. 

- It is good that the authors attempted to control for some variables (age, BMI, sex, education, and SES) in their conditional logit regression model. However, adequate information had not been given how these variables had been selected for adjustment. The decision to adjust the matching variable “age” also need to be explained. The criteria for selecting variables for adjustment should be very convincing for the readers because the association between the VD and RHD status only becomes significant after the adjustment.

RESPONSE: Thank you for pointing our attention to this. We chose the selected variables based on knowledge from previous studies about RHD and Vitamin D. We have now added references on all (ref 26-32) please see section on statistical analysis for changes. 

We chose to adjust for age also because of the later mentioned interval of age matching used in this study of 5 years. This makes a “frequency matching” on this specific variable, and thus we feel it would make sense to adjust in the regression analysis. By choosing conditional logistic regression analysis instead of unconditional, this should not be a cause of concern. We hope this explains the motivation behind the chosen variables sufficiently. 

- One of the major limitations of the study is that the vitamin D status of the study subjects was determined many years after the outcome of interest had happened. And this has not been acknowledged and discussed in the manuscript. Do you think that the current vitamin D status can be considered as a proxy indicator for the pre RHD vitamin D status?

RESPONSE: Thank you for pointing this out. Agreed, it has not been addressed properly. We acknowledge that current vitamin D status does not equate previous status. Ideally there would be data on Vitamin D concentrations on all subjects from very young age, but due to the nature of RF and RHD this would require data at least 30 years old in many cases. Unfortunately, this is not available in most areas affected by this disease, and thus we have to accept the current status as a compromise. It is not ideal, but at the same time we find it feasible as Vitamin D deficiency is dependent on sun exposure, which highly reflects both environmental and cultural factors. We reckon these factors would not have changed much over time in Nepal on an individual basis. Please see new paragraph added line 276-280 and 335-336.

- Please present the association of RHD both with Vitamin D Insufficiency and Vitamin D Deficiency so that readers could have comprehensive understanding on the relationship between the two variables. 

Thank you for noticing. The reason we did not present the OR of both vitamin D insufficiency and deficiency was the small number of people in the deficiency group (see Figure 2). We deemed the strength of the resulting OR weaker because of the small number, and thus decided to combine the two. However, we have made the adjustments as suggested. They are presented in a new Table 2 on page 13 in the manuscript for you to evaluate. 

- The sample size calculation appears to be confusing. What specific formula did you use for the determination of sample? What was the expected prevalence in the RHD subjects?

RESPONSE: Thank you the comment. We calculated sample size for two independent proportions. These proportions used were based on previous studies; according to ref 11 the expected prevalence in ARF/RHD is 77 %, and 35 % in non-RHD individuals according to ref 18. We hope this has been made clearer now in the sample size section.

- The purpose of the ROC analysis in the study is not clear and I recommend the authors to drop this analysis and related discussions and conclusions from this paper.

RESPONSE: We agree it has not been fully justified to include the ROC analysis in the manuscript. It has now been removed. 

- Please consider a sensitivity analysis to evaluate whether the decision to exclude the 12 subjects has significantly affected the conclusion of the study or not.

RESPONSE: This is a good point. The initial thought was that the 12 subjects excluded because of comorbidities would not differ on other major factors. But to see if it did have an effect, we did a sensitivity analysis of available variables by including and excluding the 12 subjects. There was no difference in major (or other) variables between the two groups and we therefore assume that the result would remain the same if we had included the 12 subjects in our model. 

- As most of the study subjects are adults, some of the variables of the study including maternal education, maternal occupation, and type of school are less relevant and I kindly recommend you to remove them. I also don’t understand the purpose of presenting HFA standard curves for boys and girls less than the age of 19 years while most of the study subjects are actually adults.

RESPONSE: This is an excellent point and we fully agree that this information is not very relevant for the adults. When designing the study we expected the mean age of patients to be lower than what was actually the case, and thus some of the questions were less relevant to the older participants. However, for wealth index score this information was still necessary which is why they were still included. We acknowledge that it does not make much sense to include in the table and have now removed the information accordingly. Instead a supplementary file of the questionnaire used including all variables have been uploaded as a supplement, as we feel it could still be useful information for others. They can then get the data collected on request if interested. 

As for the HFA: stunting is a good proxy of chronic malnutrition. Unfortunately HFA standard curves does not exceed 19 years of age and cannot be used for adults. As pointed out earlier, the majority of the study participants are adults, but we feel the knowledge about the long term nutrition, even if only available for a minority of our subjects, is still very important. Especially when taking into consideration the aforementioned limitations in study design i.e. we do not know much other about the patient’s nutritional status at the time of disease development. This is why we would like to keep the HFA on the children and adolescents we do have information on. 

- The variables of the study (apart from RHD and VD status) should be clearly listed and described in the methods section.

RESPONSE: The questionnaire with all variables can be found in the newly added supplement 1. 

- As suggested by the second reviewer, the study should be described as a case-control study. 

RESPONSE: we fully agree. It has now been corrected throughout the manuscript. 

Additional Editor Comments (section-by-section comments): 

Abstract

Is this sentence really relevant for the research question posed? “Diagnosis and treatment for Rheumatic Heart Disease (RHD) is inaccessible for many 26 of the 33 million people in low- and middle-income countries living with this disease” 

RESPONSE: Thank you for the comment. we believe so it is still important, as it sets the scene and justifies why we should investigate this disease more. 

Background

Line 94-95: Can you rephrase the phrase “occurrence of RHD” to “RHD status” because the latter is more compatible with cross-sectional and case-control designs. 

RESPONSE: Yes, we agree. It has been corrected. 

Methods

Line 118: please provide your matching criteria for age. i.e. the age range considered for matching of cases and controls

Line 154: Confidence interval > confidence level

RESPONSE: Thank you for pointing our attention to this. It has been changed and explained an age range for matching with a maximum of 5 years difference (line 121-122). 

Results

Table 1: “Less poor” > “middle”

Table 1: The variable “Type of school” is not clear

Table 1: Please redefine age ranges based on wider intervals so that statistical test would be feasible.

Line 236-237: “Analysis of the food-frequency questionnaire revealed no statistically significant differences in intake between cases and controls”. The type of the food frequency questionnaire should be described in the methods section. 

RESPONSE: Thank you. As addressed above, the intervals have been changed and the variable in question has been removed from the manuscript. Furthermore, the description is changed from fruit intake to food-frequency (line 132). We kindly refer to the questionnaire in supplementary 1 for an in-depth view of the food frequency information collected. 

Discussion

I recommend the authors to integrate the section “strength and limitation” with the discussion section.

RESPONSE: thank you for the comment, we have integrated the section into the discussion as suggested. 

Conclusion

Line 340: “Vitamin D status could not be used as a predictor of RHD diagnosis”. To start with, was there any priori hypothesis to assume that the current vitamin D status could be used as a predictor of RHD that happened many years back?

RESPONSE: We agree that we did not make that assumption from the beginning. As the ROC analysis has been removed as suggested, we have now also removed the sentence regarding this hypothesis. 

Reviewer #1: 

A well written manuscript which is clinically relevant. Please address the following :

1. Please explain the categories of the socioeconomic status : poorest, less poor and richest.

RESPONSE: We would like to refer to reference nr 25. In short SES was estimated by calculating wealth index scores for each participant using principal components analysis (PCA). PCA assigns a specific weight to each variable which is used to summarize household wealth for a specific area and divides the study population into categories based on wealth. We chose this method instead of traditional measures such as income and consumption expenditure because we find it gives a more reliable representation of households, especially in LMICs. Unreliable income patterns with short term employments is common in Nepal, and so if collecting information in a period of unemployment, results would not be representative for the actual state of the household. Measuring wealth based on durable asset ownership and access to utilities as in the PCA analysis has the advantages of providing a long-term representation of SES, not affected by fluctuations in income. It is however a relative analysis – it tells us about the composition of the studied population, but not about the national distribution. For a list of variables included please see supplementary 1. 

2. The data on hours of TV per day is redundant and sounds unscientific. Kindly remove or explain its relevance.

RESPONSE: Thank you for the comment. We have removed it from the manuscript and instead listed the variable in the supplementary as it was part of the SES calculation. 

3. Please remove the figures on height for age in girls and boys. These are not part of the study objectives.

RESPONSE: We kindly refer to the address of the same comment from the editor above. 

Reviewer #2: 

Thank you for this well designed and written study. I have only some minor revisions for the authors to consider. 

1. The title refers to both a case control and cross section study. In my view it qualifies as a case control study and so could be simply called this. It is a point in time study, yes, but this is the nature of C-C studies where we start with the known outcome and compare exposures retrospectively. The authors could reconsider the title.

RESPONSE: Thank you for the comment. We fully agree and have changed the study design to a case-control study. 

2. Minor text edits

27-28: in order to prevent and optimise treatment. Could be made clearer that it is to prevent disease and optimise treatment.

RESPONSE: We agree, this makes more sense. It has been changed as suggested. 

37-38: suggest slight change to wording with addition of 'compared with' controls instead of 'while for'

43: add s to control

77: develop-ing

RESPONSE: Thank you for good inputs. All 3 suggestions above have been added to the manuscript. 

91-92: I am not sure I understand what is meant by 'Vitamin D status vary among from 59.8%'. Could you clarify what this measurement is? e.g. low status, known status.

RESPONSE: We understand the problem with this sentence and has changed it to: Prevalence of insufficient Vitamin D levels in Nepal vary from 59.8 % amongst new mothers to 17.2 % in 6-8 year olds. We hope this makes more sense. 

116: for TB and thyroid disease, was it both current and previous disease excluded or just current?

RESPONSE: Thank you for pointing our attention to this. Subjects were excluded if they had current disease. We have changed this in the manuscript (line 119). 

119: I am interested in why congenital heart disease patients were excluded. Other studies on RHD specifically use this population for controls. Maybe a sentence to justify this would be useful for others considering designing similar studies on RHD to help them understand your decision.

RESPONSE: This is a very good point and has not been made sufficiently clear although it is important. The rationale behind excluding congenital heart disease patients were that vitamin d deficiency is more common in this group. we excluded them as they will not represent the general population in this matter. We added this explanation (line 124-125) as well as references 20-21 to support. 

Table 3: Would be good to include the % as well as in Table 1.

RESPONSE: Thank you, we have added the % as suggested. 

3. Age groups: if they were age matched, why is there a discrepancy in the age group numbers? Maybe it needs to be better explained in the methods section how you age matched so readers can understand this better.

RESPONSE: Thank you for your comment. As suggested, new age intervals have been made in table 1. The was a maximum span of 5 years between cases and controls, but usually they were the same age or only 1-2 years apart. 

4. Nutritional differences: while no differences were found among males, the respective sample sizes were small. It could then be that you simply didn't have the power to detect a true difference among the males. This could be discussed more in your limitations section.

RESPONSE: Thank you for this very good point. We agree and have added this to the discussion as: “The lack of difference in BMI and other nutritional measures amongst males should be interpreted with care, as it could simply reflect a small proportion of males included, reducing the ability to detect a difference”. 

5. 310-311: I think you need to add in references to other studies here that support your claims with regards to preponderance of females with RHD.

RESPONSE: We agree. We have added reference 28, 29, 31m that should support our claims sufficiently. 

We hope our answers have been satisfactory, and we are always ready to elaborate further if needed. 

Best Regards, 

Lene Thorup

---

## [Editor Report · Decision Letter 1]

7 May 2020

PONE-D-20-02934R1

Evaluating Vitamin D levels in Rheumatic Heart Disease Patients and Matched Controls: A Case-Control Study from Nepal

PLOS ONE

Dear Ms. Thorup,

Thank you for submitting your manuscript to PLOS ONE. After careful consideration, we feel that it has merit but does not fully meet PLOS ONE’s publication criteria as it currently stands. Therefore, we invite you to submit a revised version of the manuscript that addresses the points raised during the review process.

We would appreciate receiving your revised manuscript by Jun 21 2020 11:59PM. To enhance the reproducibility of your results, we recommend that if applicable you deposit your laboratory protocols in protocols.io, where a protocol can be assigned its own identifier (DOI) such that it can be cited independently in the future. For instructions see: http://journals.plos.org/plosone/s/submission-guidelines#loc-laboratory-protocols

We look forward to receiving your revised manuscript.

Kind regards,

Samson Gebremedhin, PhD

Academic Editor

PLOS ONE

Additional Editor Comments (if provided):

In the abstract please state the age range of the subjects.Background – line 71-73: the sentence needs citation.Background – line 87-95: please merge the two paragraphs.Methods: please clearly describe (i) how the dietary intake (Table 3) of the study subject was assessed. (ii) how the socioeconomic status was measured and classified.Results – Table 1: Please report the p-values for the age categories.Results – Line 208-11: please remove the stratified analysis by sex because the sample size is too low for meaningful sub-sample analysis.Results – Table 2: please clearly designate significant p-values in the table. All the adjusted variables must also be shown in the table.Results – Table 3: the table should have a comprehensive title that describe its content. In the methods section, the authors need to discuss the purpose of the food frequency data analysis.Results – Figure 3 and 4: As commented earlier, please remove the HFA standard curves from adolescents. The sample size (n=25) is small for any meaningful analysis. Further, as this is a comparative study, merging and describing the two groups into one is practically less meaningful.Please also discuss possible reasons that may alternative explain the association. E.g. less sunlight exposure RHD patients.Please also discuss that most of the associations are borderline and may not have huge clinical significance. Further, the implication of the small sample size of the study should be discussed.

---

## [Author Response · Author response to Decision Letter 1]

3 Jul 2020

Additional Editor Comments

• In the abstract please state the age range of the subjects.

o Thank you, the age range has been added to the abstract as suggested. 

• Background – line 71-73: the sentence needs citation.

o The treatment regime is recommended globally and described by the World Heart Federation of which the reference now has been added. The last part about recurrences is based on personal correspondence, why a reference was not added. We have chosen to omit the sentence instead, so no confusion arises. Thank you for pointing our attention to the lack of reference here. Please see changes line 70-72.

• Background – line 87-95: please merge the two paragraphs.

o Thank you. It has been changed in the final version. 

• Methods: please clearly describe (i) how the dietary intake (Table 3) of the study subject was assessed. (ii) how the socioeconomic status was measured and classified. 

o A new description has been added in the methodological section “collection of sociodemographic information”. The main difference between this study and others alike, is the use of principal components analysis, we believe. A full description of this method can be seen in ref no. 25 “Measuring Equity with Nationally Representative Wealth Quintiles Guide” (please see the Statistical analysis section), but essentially it is based what is currently owned in the household of each individual and is thus giving information of the wealth of the family on a long term basis. This method is more appropriate than basing the SES on income and expenditure information, especially in the context of Nepal, as many have fluctuating and unreliable income patterns. All variables used for the calculations can be seen in Supplement 1, questions 13-17 and 20-22 were used for the SES calculations. but to make it more clear, are also listed here:

- Own home – rented or bought.

- Household items: electricity, radio, television, mobile phone, telephone, refrigerator, bed, sofa, cupboard, computer, table, chair, clock, fan, dhiki/janto (a traditional grinding mill in Nepal). 

- Fuel used for cooking: electricity, LPG, biogas, kerosene, wood, animal dung, other.

- Type of roof on house: thatched, galvanized, ceramic, cement, other.

- No. of livestock in household: buffalo, milk cow/bull, goat, chicken, duck, pig, other. 

- Ownership of transportation equipment: Bicycle/rickshaw, motorcycle/scooter, three wheeler, car, bus or truck. 

o Similarly the fruit frequency questionnaire used can be seen in supplement 1, question no. 23. The participants ticked boxes of how many times in the last week they had consumed the mentioned items, and based on this, table 3 was created. All fruits were combined in the final table, but in the questionnaire separate fruit items were mentioned, to encourage more accurate memory/information.

Most importantly, all questions have been tailored specifically for Nepal, and have all been used in previous surveys/studies in all areas of the country.

We hope this describes the process sufficiently, and likewise that you will contact us with further questions if not so.

• Results – Table 1: Please report the p-values for the age categories.

o Thank you, it has been added to table 1. 

• Results – Line 208-11: please remove the stratified analysis by sex because the sample size is too low for meaningful sub-sample analysis.

o Thank you for pointing this out. The sub-sample analysis have been removed from the result section as asked, as well as from Table 1. 

• Results – Table 2: please clearly designate significant p-values in the table. All the adjusted variables must also be shown in the table.

o We have marked the statistically significant p-values with an “*” and listed the adjusted variables in the Table text beneath marked with an “a”. We hope this is satisfactory and gives a clear overview of the table content. 

• Results – Table 3: the table should have a comprehensive title that describe its content. In the methods section, the authors need to discuss the purpose of the food frequency data analysis. 

o Thank you for the comment. We agree a purpose is justified. We have added an explanation in the methods section line 134 – 141 elaborating on the purpose behind this questionnaire. To summarize we wanted to characterize the general quality and variance in food consumption in the two groups, as well as investigate whether a difference in vitamin D rich foods were detectable and in concordance with the measured vitamin D concentrations. 

• Results – Figure 3 and 4: As commented earlier, please remove the HFA standard curves from adolescents. The sample size (n=25) is small for any meaningful analysis. Further, as this is a comparative study, merging and describing the two groups into one is practically less meaningful.

o After careful consideration, we have come to the same conclusion and removed the section on HFA standard curves from the manuscript. Both in the result section and the discussion. Initially we found the finding quite interesting, but as you have pointed out, the sample size is simply to small. We will keep the finding in mind for possible further investigations. Thank you for making a very good and important point. 

• Please also discuss possible reasons that may alternative explain the association. E.g. less sunlight exposure RHD patients.

o Thank you for the comment. We have chosen to incorporate it as a possible explanation for the lower concentrations in the patients and something to look further into for future studies. Please see the discussion line 265-267. 

• Please also discuss that most of the associations are borderline and may not have huge clinical significance. Further, the implication of the small sample size of the study should be discussed.

o We understand these very relevant points. A section has been added to the manuscript under the Discussion section line 320-324. We would like the reader to understand that the small association might be explained by the smaller sample size, but of course a larger study might turn out dismissing the associations completely. Outcomes only future studies could clarify.

---

## [Editor Report · Decision Letter 2]

6 Aug 2020

Evaluating Vitamin D levels in Rheumatic Heart Disease Patients and Matched Controls: A Case-Control Study from Nepal

PONE-D-20-02934R2

Dear Dr. Thorup,

We’re pleased to inform you that your manuscript has been judged scientifically suitable for publication and will be formally accepted for publication once it meets all outstanding technical requirements.

Kind regards,

Samson Gebremedhin, PhD

Academic Editor

PLOS ONE
---

## [Editor Report · Acceptance letter]

12 Aug 2020

PONE-D-20-02934R2 

Evaluating Vitamin D levels in Rheumatic Heart Disease Patients and Matched Controls: A Case-Control Study from Nepal 

Dear Dr. Thorup:

I'm pleased to inform you that your manuscript has been deemed suitable for publication in PLOS ONE. Congratulations! Your manuscript is now with our production department. 

Kind regards, 

on behalf of

Dr. Samson Gebremedhin 

Academic Editor

PLOS ONE